# Uptake and use of a minimum data set (MDS) for older people living and dying in care homes in England: a realist review protocol

Massirfufulay Kpehe Musa ![ORCID],[1,2] Gizdem Akdur ![ORCID],[2] Barbara Hanratty ![ORCID],[3,4] Sarah Kelly ![ORCID],[5] Adam Gordon ![ORCID],[6,7] Guy Peryer ![ORCID],[8] Karen Spilsbury ![ORCID],[9,10] Anne Killett,[8] Jennifer Burton ![ORCID],[11] Julienne Meyer,[12] Sue Fortescue,[13] Ann-Marie Towers ![ORCID],[14,15] Lisa Irvine ![ORCID],[2] Claire Goodman ![ORCID][2,16]

► Prepublication history and additional materials for this paper is available online. To view these files, please visit the journal online (http://dx.doi.org/10.1136/bmjopen-2020-040397).

For numbered affiliations see end of article.

**Correspondence to**
Dr Claire Goodman;
c.goodman@herts.ac.uk

## ABSTRACT

**Introduction** Care homes provide nursing and social care for older people who can no longer live independently at home. In the UK, there is no consistent approach to how information about residents' medical history, care needs and preferences are collected and shared. This limits opportunities to understand the care home population, have a systematic approach to assessment and documentation of care, identifiy care home residents at risk of deterioration and review care. Countries with standardised approaches to residents' assessment, care planning and review (eg, minimum data sets (MDS)) use the data to understand the care home population, guide resource allocation, monitor services delivery and for research. The aim of this realist review is to develop a theory-driven understanding of how care home staff implement and use MDS to plan and deliver care of individual residents.

**Methods and analysis** A realist review will be conducted in three research stages.
Stage 1 will scope the literature and develop candidate programme theories of what ensures effective uptake and sustained implementation of an MDS.
Stage2 will test and refine these theories through further iterative searches of the evidence from the literature to establish how effective uptake of an MDS can be achieved.
Stage 3 will consult with relevant stakeholders to test or refine the programme theory (theories) of how an MDS works at the resident level of care for different stakeholders and in what circumstances. Data synthesis will use realist logic to align data from each eligible article with possible context–mechanism–outcome configurations or specific elements that answer the research questions.

**Ethics and dissemination** The University of Hertfordshire Ethics Committee has approved this study (HSK/SF/UH/04169). Findings will be disseminated through briefings with stakeholders, conference presentations, a national consultation on the use of an MDS in UK long-term care settings, publications in peer-reviewed journals and in print and social media publications accessible to residents, relatives and care home staff.

### Strengths and limitations of this study

► The review will identify what needs to be in place to support the implementation of an minimum data sets (MDS) in long-term care settings where standardised approaches to resident assessment and data collection are not in place.

► The review will demonstrate how using an MDS affects the everyday work and care practices of staff and its impact on residents' care.

► The synthesis will integrate qualitative and quantitative evidence that offers transferable learning for long-term care settings that do not currently use an MDS.

► There are time constraints that may result in the team focusing on or prioritising some aspects of an MDS implementation over others.

**PROSPERO registration number** CRD42020171323; this review protocol is registered on the International Prospective Register of Systematic Reviews.

## BACKGROUND

There are nearly 12 million (11 989 322) people aged 65 years and above in the UK of which an estimated 1.6 million are aged 85 years and above, and more than 500 000 (579 776) people are aged 90 years and above.[1] However, with greater longevity (eg, age 85 years and above) comes higher levels of dependency, dementia and comorbidity,[2] which in turn intensify the need for social care services.[3] Approximately 420 000 older people in England and Wales live in care homes.[4] Care home is a generic term that refers to facilities where a number of older people live together and have staff available 24 hours to provide personal care (eg, residential care or assisted living/supportive housing facilities), and for those facilities

where a qualified nurse is required on duty 24 hours to provide additional nursing care for more dependent residents (eg, nursing homes or skilled nursing facilities).[5] The care home population represents the oldest and most vulnerable group of older people,[6] with approximately 70% of them living with cognitive impairment[6 7] and 76% requiring assistance with mobility.[8]

In the UK, there is no consistent approach to how information about residents' medical history, care needs and preferences is collected and used. The absence of a national mandate, lack of links with National Health Services (NHS) data and implementation challenges have meant that a minimum data set (MDS) and data-driven approaches to resident assessment have been limited to single projects.[9] The lack of a link between care home data and the NHS data recently became evident when figures reported by the Office for National Statistics during the first three weeks of COVID-19 underestimated the impact of the pandemic among care home residents.[10] All care homes, however, routinely collect large amounts of data about their residents. The challenge is to establish systems of assessment and recording that are evidence-based, accessible and valuable to those using, providing, commissioning and regulating care services. Without a unified record, there may be duplication of assessments, communication failures and unmet care needs.[11 12] Determining consistent ways to assess and document care for residents in the care home settings is a priority because over the next two decades, the number of older people likely to need long-term care will increase.[2]

Minimum Data Set in this realist review is defined as a comprehensive, standardised account of the characteristics and needs and ongoing care of residents living in long-term care (care home) settings. The review concentrates on how MDS is used by care home staff and what supports effective uptake for the benefit of individual residents. It also takes account of the involvement of residents themselves and their family in influencing how an MDS is used.

There are multiple versions of MDS, which are often country specific. However, all versions of MDS share a common language and are designed to support an integrated system of care that can support cross-sector clinical and managerial decision-making. One example of an MDS is the International Resident Assessment Instrument (Inter-RAI), which is developed for long-term care facilities.[13] The use of an MDS is often mandated and/or linked to national reimbursement systems and quality monitoring. Research has demonstrated the value of an MDS to commissioners and service providers in enabling identification of care needs and residents at risk of ill health.[14–18] They can provide a comprehensive account of resident characteristics, resource use and care outcomes in key areas (eg, activities of daily living, cognitive performance, pain, cost of care and infection).[19] However, Kontos and colleagues argue that a standardised process such as the MDS fails to consistently result in individualised care planning, which may suggest problems with content of an MDS.[20]

There is a dearth of information on For long-term care settings making the transition to standardised approaches to data collection, what needs to be in place to implement an MDS and how its use impacts on staff work, time away from care, knowledge of the care home residents, working with other healthcare professionals and benefits (or not) to residents, staff and residents' families.

## Review aim and objectives
### Aim
To develop a theory-driven understanding of how care homes' staff can effectively implement and use a Minimum Data Set (MDS) to plan and deliver care of individual residents.

### Objectives
1. Develop a programme theory describing contexts that can support the uptake and use of an MDS in care homes.
2. Identify in what circumstances the use of an MDS produces improved outcomes (including resource use) for an individual resident, their family and the care home staff and their employing organisation.

## Study design
We will conduct a realist review that seeks to formulate, test and refine the programme theory while assessing whether and how the programme succeeds in the local setting,[21] in order to generate important insights into the UK. A programme theory is an overarching theory or model of how a programme, or an intervention, is expected to work[22] and it helps to explain (some of) 'how and why, in the 'real world', a specific programme 'works', for whom, to what extent and in which contexts'.[23] The unit of analysis in a realist review is the ideas and assumptions (ie, the programme theories) that underlie an intervention and explain how it works to achieve the desired outcomes.

A realist review is an interpretive, theory-driven approach to evidence synthesis[24 25] to develop a programme theory of the causal processes and context-specific factors that can explain how an intervention or programme is expected to work. Realism is a methodological paradigm that sits between positivism (the world is real and can be observed directly) and constructivism (given that all we know has been processed through the human mind, we can never be sure exactly what reality is).[26] It is flexible to changes and embedded in a social reality that influences how a programme is implemented and how various actors in that reality respond to it.[21]

Programmes like the MDS will always rely on human agency to affect change. A realist approach argues that the features or elements of the programme will produce a range of potential responses to the programme which will impact on the outcomes.[21 24] It assumes that there is a knowable, independent reality that will shape how different participants react to a programme, whether they are aware of these influences or not.[27] Thus, uptake and implementation of an MDS can lead to different outcomes for different stakeholders (eg, residents and their relatives, staff, commissioners, regulators) depending on who

is involved, the resources available and how the MDS is used.[28 29]

Using a realist approach,[21] there are four key linked concepts for explaining and building a theory of how a programme works: (1) contexts (C), which are often the 'backdrop' of interventions[24], (2) mechanisms (M), which are not observed directly but account for what it is about programmes that make them work,[30 31] characterised as 'a process that bring about or prevents some change in a concrete system'[32] and (3) outcomes (O) of the intervention (planned or unplanned, visible or not) or strategies of the intervention.[33] It is the context–mechanism–outcome (CMO) configurations (models indicating how programmes activate mechanisms for whom and in what conditions, to elicit outcomes) that are the building blocks of the theory.[25] Thus, in care home settings, staff understanding of their responsibility for completing an MDS could be a context (C), which triggers how staff prioritise recording information as part of care work (M) to identify residents at risk of deterioration (O).

The review will follow Pawson's five practical stages of conducting realist reviews: clarify the scope of the review, search for evidence, appraise primary studies and extract data, synthesise the evidence and draw conclusions and disseminate the findings.[21] Organised in three stages, we will first undertake a scoping of the literature to identify care home-specific work on the uptake of MDS and develop relevant theories around staff uptake and implementation and outcomes specific to the use of an MDS in long-term (care home) settings. Stage 2 will test and refine the emergent theories that underpin the use of an MDS and that leads to both intended and unintended outcomes for staff and residents. Stage 3 will synthesise the findings to establish how and when the use of an MDS achieves different outcomes for residents, families, staff and organisations.

### Stages of the review process
#### Stage 1: Defining the scope of the review, identifying existing theories and theory development
This review is nested within a larger review ('A systematic review of process and contextual factors that influence research implementation in care homes and identification of key measures and outcomes in care home research'). The literature identified from the larger review will be the starting point for the scoping work.

The scoping of the literature will focus on studies that report on how an MDS is used in long-term care (care home) settings. Outcomes of interest will be established by the project team as an iterative process but are likely to include evidence of its impact on: accuracy of reporting, needs assessment, staff workload, quality of care, resource use, staff satisfaction and access to care.

#### *Literature search strategy*
Searches for relevant evidence will include databases of peer-reviewed literature Medical Literature Analysis and Retrieval System Online (MEDLINE), Excerpta Medica dataBASE (EMBASE), Cumulative Index of Nursing and Allied Health Literatur (CINAHL), Applied Social Sciences Citation Index and Abstracts and sources of the grey literature (including OpenGrey and websites of organisations relevant to care homes and care of older people). Studies for inclusion will be limited to English language. We will search data from January 2009 to March 2020. These initial searches will be complemented by:

1. Searching of both lateral and forward citations of included papers paying particular attention to seminal papers on the uptake and use of an MDS in long-term care settings.
2. Contact with experts who have developed and/or use an MDS.

The search strategy will be iterative because predetermined linear search strategies are unlikely to generate search results that are adequate for purposes of conducting knowledge-building and theory-generating reviews.[34] Throughout the proposed review and based on the scoping review findings, we will introduce new, targeted search terms not defined in the initial searches.[34 35] We will use search terms such as care homes, skilled nursing facilities and nursing homes (see online supplemental table S1). We will then combine these terms with other terms such as MDS, inter-RAI, Research Assessment Instrument and RAI using Boolean logic (see online supplemental table S1). A comprehensive list of search terms and databases used will be provided in subsequent publications on completion of the proposed research.

#### *Literature screening process*
Search results relevant to MDS will be downloaded into Covidence software. Screening and selection of articles will take place in two stages (title and abstract, and full text).[36] Two reviewers (MKM and GA) will independently screen titles and abstracts identified by electronic search and applied the selection criteria to potentially relevant full-text papers.[37] The two reviewers (MKM and GA) will then independently screen 10 articles and cross-check results to discuss emergent ideas and themes and establish consensus on the relevance of the documents. Disagreement between MKM and GA will be resolved by the third reviewer, CG.

Based on earlier work that used an MDS to collect data and cross team discussions,[38] the initial programme theory will focus on how an MDS is used in long-term care. This will be the basis for scoping the literature on the challenges of changing systems of care, the need for a policy or regulator mandate, how it affects patterns of working in the care home, staff involvement in data entry and changes in residents' care. At this stage, we will not be assuming causality but we will recognise that these are likely to influence uptake and use and resident and staff outcomes. Studies have used MDS, or similar approaches to document resident, staff and organisational outcomes but do not address questions of implementation and use will be reviewed to identify supplementary evidence on related issues of interest (eg, accuracy of data, time

commitment and how information was used and by whom). The search strategy will include citation searching and grey literature and will be iteratively extended and refocused as the review progresses.

### Formulating initial programme theories

At this stage, we will investigate demi-regularities in outcome patterns by developing a series of 'if–then statements' from the scoping literature to summarise the dominant arguments and supporting evidence of what supports the uptake and use of MDS in long-term care settings. This will inform the development of hypotheses that posit possible contexts (C) that are the backdrop to successful (or not) uptake[24] of MDS; the mechanisms (M) they trigger[32] and planned or unplanned outcomes (O) arising from the use of MDS.[33] Possible CMO configurations will be discussed across the research team and with subject experts on MDS and will inform stage 2 of the review phase and additional searches.

There are several ways to conceptualise the development and uptake of MDS as many have their roots in medical approaches to assessment and health systems design. It is therefore likely that the review will be informed by and aimed to build on theories of implementation in long-term care,[32 39] uptake of technological innovation[40 41] assessment of older people with complex needs,[2 4 8] person-centred care[42 43] and risk management and quality assurance.[44]

The introduction of an MDS in care homes is sensitive to the resource and policy constraints under which the care homes operate. Candidate theories will therefore consider the role of the regulator and legal frameworks that incentivise (or not) data sharing across organisations.

### Literature selection, quality appraisal and data extraction criteria

There will be no restriction on the types of study design for eligibility.[45] Article selection will be based on the extent to which research on the uptake and routine use of an MDS can contribute to the development of a programme theory of implementation of MDS in long-term care settings.

Included studies are likely to cover the following:

► Studies on the introduction and development of an MDS with care home staff.
► Studies that focus on the inclusion and engagement of care home staff, residents and their representatives in sharing resident data with the specific remit of improving resident outcomes.
► Implementation studies that provide evidence on what facilitates and inhibits the shared documentation and care planning in care home setting including digital innovation.
► Studies on commissioning services for care homes based on care home-generated data.

No geographical restrictions will apply, although we will only include studies that are published in English and focus on the uptake and use of an MDS in long-term care settings.

### Quality appraisal of included articles

The quality of included papers will be carried out in accordance with previous appraisal work within a realist project.[29 36] The quality appraisal of included studies will be combined with data extraction, a technique usually employed in realist review.[46] Realist reviews employ various techniques to assess the quality of evidence by drawing on evidence from a wider range of sources unlike traditional systematic reviews that only focus on the methodological quality of studies.[29 36] As quality of evidence is not limited to the methodological quality, or hierarchy of evidence in realist reviews,[29] each article in this review will be assessed based on its trustworthiness and applicability to the research questions. Consistent with the realist approach, we will use an iterative approach to determine whether an article is considered 'good enough and relevant' to answer our research questions.[47] Good enough will be based on the reviewers' own assessment of the quality of evidence, for example, if it is considered to be of a sufficient standard for the research question, and relevance will relate to whether the authors provided sufficient descriptive detail and/or theoretical discussion to contribute to the initial programme theories development.[37]

The quality appraisal in this review will be assessed on a case by case basis considering the opportunities for learning, scientific rigour of evidence and relevance to the review questions.[46] Two reviewers (MKM and GA), in consultation with CG, will lead this process. Weaker papers and those with equivocal or negative findings will be considered if they contribute to the overall programme theories.

### Data extraction

Data extraction will be conducted on the basis of relevance to the review questions and will be based on realist guidelines to address questions that explore 'what is it that supports (or hinders) an MDS implementation in care homes, and how care home staff use and interpret an MDS to guide residents' care?'[36] From the extracted data, two reviewers (MM and GA) will independently rate the studies as either yes, no or maybe in terms of whether the particular article meets inclusion criteria. We will use 'maybe' for issues that cannot be answered based on the information available in the publication. Then the two reviewers (MM and GA) will meet with a third reviewer (CG) who will serve as an advisor to verify, confirm or reject inclusion of the data. From relevant articles, several 'if–then' statements will be made from which initial programme theories will be made.

### Data synthesis

Data synthesis of the scoping phase will use realist logic of analysis to align data from each eligible article with possible CMO configurations[21] or specific elements that answer the research questions. These emerging findings and putative patterns of association within the data will be tested further in stage 2 to build causal explanations based on the observed interactions between CMO.

**Box 1    Stakeholders' consultation**

During stakeholders' consultation interviews, we will explore:
1. The fit between the emerging programme theories and how stakeholders understand what is needed for the development and use of minimum data set (MDS) in long-term care settings.
2. Alternative explanations stakeholders identify as relevant for the successful use of MDS by care home staff.

## Stage 2: Testing and refining the programme theories

Further iterative searches of the evidence will be directly informed by the CMOs developed in stage 1 as candidate programme theories. The iterative circle will continue throughout the course of the review until theoretical saturation has been achieved.[21 48]

Data will be extracted using a bespoke data extraction form. It will include descriptive data on study characteristics and is likely to focus on what can be learnt about the role and work of staff, resources required to implement an MDS, features of the settings (eg, workforce capacity, size of care homes), explicit and implicit theories for how interventions were anticipated to work and patient and carer outcomes. A sample of the papers, including those that appear to offer most learning, and their completed data extraction forms will be shared across the project team to support ongoing discussion and debate of the candidate theory(ies) and their supporting evidence.

## Stage 3: Analysis and synthesis of evidence from the proposed programme theories

To support hypothesis refinement and 'fine tune' the theory(ies) that show the most promise, we will further test findings from stage 2 in a series of interviews.[49] We will do this through stakeholders' consultation (box 1). It is acknowledged in realist research that published literature alone cannot help to unearth the reasoning of end users of a programme.[21 26 31] Therefore, the inclusion of primary data from stakeholders in the review will be an added value.

We will carry out up to eight individual semistructured interviews with frontline staff (staff from care homes who use predominately paper-based records and staff who routinely use electronic records for their residents) and care home managers and stakeholders who are experts regarding the use of care home residents' data. The semistructured interviews will be guided by emerging programme theories from the early stages of the review.

Participants' selection will be purposive based on their knowledge of using MDS in and with care homes. All participants will be sent a detailed participant information sheet via email and consent form prior to the interview. Interviews will be either face-to-face, online or telephone conversations. Interviews will be audio recorded and transcribed.[21] Data will help to refine or refute the demi-regularities seen in outcome patterns emerging from the empirical literature.[26]

At the end of the interviews, we will present and discuss the programme theories, with the supporting evidence for discussion, with the whole research project team.

A summary of the review process is presented in figure 1. The double arrows within or between stages indicate iterative processes of the review.

The final programme theories will be synthesised narratively, by logic models and/or summary tables where appropriate. The findings of the review will be written up according to the Realist And Meta-narrative Evidence Syntheses: Evolving Standards (RAMESES) guidance.[48]

## Patient and public involvement

To keep the person being cared for at the centre of our thinking in ways that inform delivery of care or care home resident benefit, we will convene two care home-based resident patient and public involvement (PPI) groups that will meet throughout this review project. A member of this realist review, who is a former carer and IT specialist, will lead the PPI groups. We anticipate that input from residents and carers will help us to identify and understand the important contextual factors, the resource and reasoning that support the implementation of an MDS in long-term care settings. The PPI groups input will help us to tailor the stakeholders' consultation, inform our theory (or theories) development and ensure that the final refined programme theory(ies) resonates with care home staff and residents' experience.

## DISCUSSION

This realist review will provide a theory-driven understanding of what needs to be in place for the successful implementation an MDS systems in care home settings

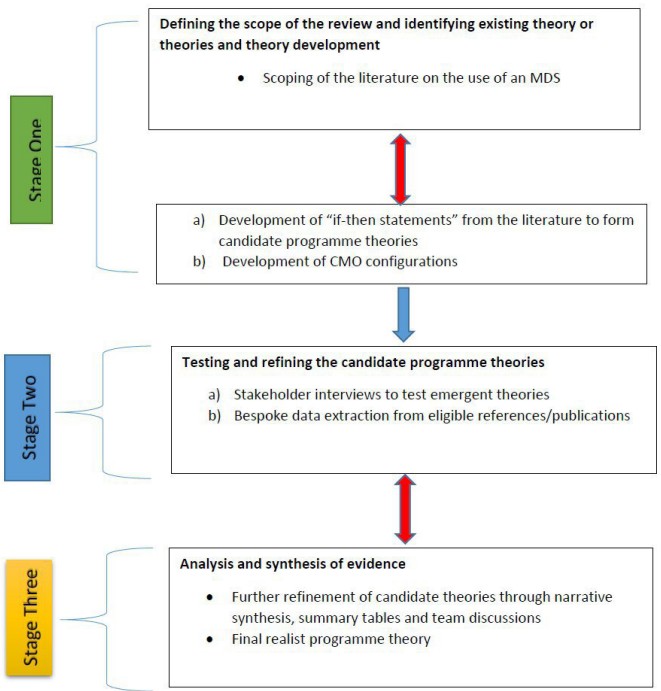

**Figure 1**    The realist review processes. CMO, context–mechanism–outcome; MDS, minimum data set.

to benefit residents, staff, families, service managers and commissioners.

Research has demonstrated the value of MDS to commissioners and service providers in the identification of care needs.[50–54] A research study that used care home-specific MDS identified particular implementation challenges.[38] It enabled comprehensive analysis of baseline resident data and residents at risk but there was limited staff capacity to support and sustain its completion over time when it was not linked to other data collection responsibilities.[38] This review addresses a gap in the evidence about what is needed to support uptake and implementation of an MDS, what needs to be in place for effective uptake and how an MDS is used in different circumstances to enable key care outcomes for residents. By identifying the causal mechanisms at work, the review findings will directly inform decision-making about how to design, tailor and implement an MDS that is acceptable to staff and can inform residents' everyday care.

## ETHICS AND DISSEMINATION

The University of Hertfordshire Ethics Committee has approved this study (HSK/SF/UH/04169). Findings will be disseminated through briefings with stakeholders, conference presentations, a national consultation on the use of an MDS in UK long-term care settings and publications in peer-reviewed journals and in print and social media publications accessible to residents, relatives and care home staff.

**Author affiliations**
¹Faculty of Nursing, Midwifery and Palliative Care, King's College London, London, UK
²Centre for Research in Public health and Community Care (CRIPACC), School of Health and Social Work, University of Hertfordshire, Hatfield, United Kingdom
³Population Health Sciences Institute, Campus for Ageing and Vitality, Newcastle University, Newcastle upon Tyne, United Kingdom
⁴NIHR Applied Research Collaboration, North East and North Cumbra, UK
⁵Institute of Public Health, University of Cambridge, Cambridge, UK
⁶Division of Rehabilitation, Ageing and Wellbeing, School of Medicine, University of Nottingham, Nottingham, United Kingdom
⁷NIHR Applied Research Collaboration, East Midlands, UK
⁸Faculty of Medicine and Health Sciences, University of East Anglia, Norwich, UK
⁹School of Healthcare, University of Leeds, Leeds, UK
¹⁰NIHR Applied Research Collaboration, Yorkshire and Humber, UK
¹¹Institute of Cardiovascular and Medical Sciences, University of Glasgow, Glasgow, UK
¹²National Care Forum/Care for Older People, School of Health Sciences, Division of Nursing, City, University of London, London, United Kingdom
¹³Alzheimer Society Research Network, London, UK
¹⁴Centre for Health Services Studies, University of Kent, Canterbury, UK
¹⁵NIHR Applied Research Collaboration, Kent Surrey and Sussex, UK
¹⁶NIHR Applied Research Collaboration, East of England, UK

**Acknowledgements** Arne Wolters, Liz Jones and Iain Lang are collaborators on the Developing research resources And minimum data set for Care Homes' Adoption and use (DACHA) study and contributed to early conversations about the review and lead work in the larger study that will use these findings. We thank the reviewers Catherine Powell and Matthias Hoben whose constructive comments/suggestions helped to improve and clarify this manuscript.

**Contributors** Concept and design of the review are embedded in the Developing research resources And minimum data set for Care Homes' Adoption and use (DACHA) study. SK conducted the initial literature search and MKM, GA and CG wrote the first draft of the manuscript. Critical review and refinement of the manuscript were provided by GP, KS, AK, JB, BH, AG, JM, SF, A-MT and LI. MKM, GA and CG approved the final version.

**Funding** This study/project is funded by the National Institute for Health Research (NIHR) Health Service Research and Delivery programme (HS&DR NIHR127234) and supported by the NIHR Applied Research Collaboration (ARC) East of England. This review started from December 2019 and it will end by 31st December 2020.CG is a NIHR Senior Investigator

**Disclaimer** The views expressed are those of the author(s) and not necessarily those of the NIHR or the Department of Health and Social Care.

**Competing interests** None declared.

**Patient consent for publication** Not required.

**Provenance and peer review** Not commissioned; externally peer reviewed.

**ORCID iDs**
Massirfufulay Kpehe Musa http://orcid.org/0000-0001-5506-4720
Gizdem Akdur http://orcid.org/0000-0001-7326-4750
Barbara Hanratty http://orcid.org/0000-0002-3122-7190
Sarah Kelly http://orcid.org/0000-0002-1114-2456
Adam Gordon http://orcid.org/0000-0003-1676-9853
Guy Peryer http://orcid.org/0000-0003-0425-6911
Karen Spilsbury http://orcid.org/0000-0002-6908-0032
Jennifer Burton http://orcid.org/0000-0002-4752-6988
Ann-Marie Towers http://orcid.org/0000-0003-3597-1061
Lisa Irvine http://orcid.org/0000-0003-1936-3584
Claire Goodman http://orcid.org/0000-0002-8938-4893

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
