## [Reviewer comments · BMJ Open]

ARTICLE DETAILS

TITLE (PROVISIONAL)	The uptake and use of a minimum data set (MDS) for older people living and dying in care homes in England – a realist review protocol
AUTHORS	Musa, Massirfulay; Akdur, Gizdem; Hanratty, Barbara; Kelly, Sarah; Gordon, Adam; Peryer, Guy; Spilsbury, Karen; Killett, Anne; Burton, Jennifer; Meyer, Julienne; Fortescue, Sue; Towers, Ann-Marie; Irvine, Lisa; Goodman, Claire

VERSION 1 – REVIEW

REVIEWER	Catherine Powell University of Bradford, UK
REVIEW RETURNED	04-Jun-2020

GENERAL COMMENTS	1. Is the research question or study objective clearly defined? Yes the study objective is very clearly defined. (To develop a theory-driven understanding of how internationally deployed Minimum Data Sets (MDS) are implemented in care home settings and how their use by care home staff influences outcomes for residents, family, staff and linked organisations.) 2. Is the abstract accurate, balanced and complete? The abstract is well written with relevant sections completed. BMJ Open requires authors of protocol papers to report dates of the study in the manuscript. There is a registration start date of 09/03/2020, but there is no end date indicated. The objective 'To offer transferrable learning and/or utility for an MDS development in countries that do not have an MDS for older people in long term care settings' needs to be clearer in the abstract. 3. Is the study design appropriate to answer the research question? Taking a realist approach is very well suited to meeting the research objective which is theory driven, seeking to gain a deeper understanding of implementation and use in care homes; settings which can be diverse. 4. Are the methods described sufficiently to allow the study to be repeated? Methods are clear and structured by stages of the review process. Authors describe how another review and feeds into this review. 5. Are research ethics (e.g. participant consent, ethics approval) addressed appropriately? From the statement 'The protocol has been submitted to the University of Hertfordshire Ethics Committee.'-does this means the research is currently under ethical review and not yet approved? Stage 3 mentions a series of interviews. The participant consent process needs to be explained. 6. Are the outcomes clearly defined?
---

	N/A 7. If statistics are used are they appropriate and described fully? N/A 8. Are the references up-to-date and appropriate? Yes references are up to date and appropriate to a realist review in care homes. 9. Do the results address the research question or objective? N/A 10. Are they presented clearly? N/A 11. Are the discussion and conclusions justified by the results N/A 12. Are the study limitations discussed adequately? Yes limitations are discussed by the authors. 13. Is the supplementary reporting complete (e.g. trial registration; funding details; CONSORT, STROBE or PRISMA checklist)? Review registration number is provided. PRISMA-P might have been used, however as this is a realist review its acceptable that its not been provided. 14. To the best of your knowledge is the paper free from concerns over publication ethics (e.g. plagiarism, redundant publication, undeclared conflicts of interest)? There does not appear to be any ethical issues. 15. Is the standard of written English acceptable for publication? Standard of English is acceptable.
--	---

REVIEWER	Matthias Hoben University of Alberta, Canada
REVIEW RETURNED	19-Jul-2020

GENERAL COMMENTS	The authors propose to conduct an important review – a realist review focusing on better understanding the implementation of minimum data sets (MDS) in residential elder care settings. Population-based measurement and monitoring of quality of care is critical to detect and act upon issues. Therefore, this review addresses an important topic, especially given the fact that such population-based monitoring currently is not done in care homes in the UK. However, there are various major issues with this protocol that I will outline in the following: Clarification and consistency are needed on what the actual focus of the review is. First, there is no clear aim/objectives statement in the abstract. Elements of what should go in such a statement are included in the introduction section of the abstract (suggests that there is a gap related to our understanding of how MDS are implemented and how different target groups use and interpret MDS data – and raises the impression the review will address this gap). Then the first sentence in the methods section of the abstract suggests that the review aims to better understand, what supports the implementation and use of an MDS. Better understanding (a) how MDS are implemented (i.e. which implementation strategies are used), (b) how information of implemented MDS are used and interpreted and (c) factors that support or hinder implementation and use of an MDS (barriers and facilitators) are three very different aims. If the review wants to address all three, this should be made explicit – in one location of the abstract. If the review wants to address only one or some of them, this should be made clear too – and the other statements should be removed. Second, the strengths and limitations box adds a fourth possible review focus: (d) how, why and under what circumstances does the implementation of an
--

MDS lead to benefits (effectiveness, possible side effects and their reasons/mechanisms). Not only is that a focus not yet mentioned, it actually contradicts the statement in the introduction section of the background suggesting that we already know that MDS “help to improve residents’ outcomes and inform care commissioning”. Third, additional gaps and possible foci are mentioned in the background (lines 96-100): (e) How does using MDS affect the everyday work and care practices of staff, and (f) how are different priorities of various groups of persons met? Also, (g) what resources are required to sustain use of MDS? The objective statement and questions at the beginning of the methods section again communicates a slightly different focus. I suggest the authors clearly define what they want to achieve with this review, clearly state it and communicate this consistently throughout the paper (including abstract, strengths/limitations box, main manuscript text).

Background

I understand that BMJ Open is a journal that is situated in the UK and that in the UK there is probably a common understanding of what a care home is. However, I also understand that BMJ Open targets an international audience – and internationally, there is a huge heterogeneity with respect to terminology used to describe congregate care settings (nursing homes, care homes, LTC sites, retirement homes, assisted living, supportive housing, etc. – see this article: [https://www.jamda.com/article/S1525-8610\(14\)00838-X/fulltext](https://www.jamda.com/article/S1525-8610(14)00838-X/fulltext). I think I understand that there are two types of care homes in the UK: residential care homes and nursing care homes. the latter serve residents with more complex care needs, provide 24-h nursing care by nurses hired by the site, while residential care homes serve residents with less complex care needs and care is primarily provided by care assistants. Please define clearly (and if possible in an international context) what you mean by care homes. For example, when compared to the situation in North America, it sounds like residential care homes are more like what is called assisted living or supportive housing there, while nursing care homes are closer to what is called nursing homes or residential LTC in North America.

Lines 83/84: The sentence “Over the next two decades older people likely to need long term care will increase” needs to be fixed. Something is missing (the number of older people?).

Lines 84/85: I think it should be “is a priority”, rather than “are”. Determining X is a priority ... and X is “consistent ways to assess and document”. Ways is plural but you are describing one priority (determining these ways), rather than multiple priorities.

Line 87: Either say “the use of MDS has been adopted” (since use is singular) or “MDS have been adopted” (because then MDS is plural). Right now you are mixing plural and singular.

Line 88/89: same issue as in my previous comment: one international consortium (singular) of researchers and practitioners has developed ... or researchers and practitioners (plural) have developed ...

I think the authors need to discuss in more detail which different versions of an MDS they are interested in. They only mention the interRAI LTCF version. However, the earliest version of an MDS that was implemented system-wide was the RAI-MDS 1.0 (1990 in the

US). That version was updated to the version 2.0, which was implemented in the mid 1990s. Then, since 2010 the USA uses the MDS 3.0. In Canada most provinces use the MDS 2.0, some have started to experiment with the interRAI LTCF version. Apart from the RAI family of tools, are the authors interested in other tools? For example PLAISIR – a tool developed and used in the French-speaking part of Canada and used in some regions in Switzerland? Another tool used in Switzerland as an alternative to the RAI-MDS is BESA. There may be more. I think the authors need to be clear on what types of tools they are interested in – and if their interest goes beyond the RAI family of tools, they need to say so and discuss those tools too. Since the characteristics of a tool are an important set of factors that influence implementation, these considerations will be critical for their review.

Methods

The authors rightfully state that realist reviews are “theory-driven”. Given that, the discussion on what their exact initial theoretical considerations are is insufficient. Just stating that the realist review will be embedded within the context of a larger systematic review and then list various possible theoretical perspectives that may inform the review (lines 176-184) is not enough. Very likely this very broad and superficial approach of discussing candidate theories may be rooted in the fact that the authors have not clearly and consistently stated the exact aim and objectives of the review. If the objectives and questions stated in the methods section indeed are what will guide the review, this needs to be reflected in the other sections of the protocol – and consistently so. Right now it appears that the authors’ understanding of the review focus is a bit fussy and varies.

The authors talk about the development of a programme theory – but nowhere do they define exactly what that means. There are excellent definitions of programme theories in the realist methods literature and I highly encourage the authors to clearly define what they mean by ‘programme theory’, using this literature. Also, even if development of a programme theory is the aim of the review, usually starting with an initial programme theory that then is revised, refined, etc. – at least some idea of what the theoretical considerations may be – is critical.

Included studies: it sounds like the authors will largely focus on studies on care staff experiences with MDS implementation or studies that focus on facility-level implementations and related factors. However, we know that system factors and approaches (legislation, system support, resources, incentives, etc.) are as important for implementation success – especially if a system-wide implementation is the ultimate goal.

The information on how study quality will be determined is insufficient. Just referring to and citing a previous realist review is not enough. Details on how the team will assess whether a paper is ‘weak’ or ‘strong’ and whether it is ‘relevant’ or not are needed.

The entire protocol seems to be driven by the assumption that it is already established that implementing MDS generally has benefits and that it is really just about understanding how to best implement these tools and use their data. However I think the authors have overlooked important issues in this context: possible negative side effects of implementing and using MDS

	(https://www.ncbi.nlm.nih.gov/pmc/articles/PMC2867498/, https://utpjournals-press.login.ezproxy.library.ualberta.ca/doi/full/10.3138/jcs.50.2.348), differences in perspectives between decision makers/scientists who may think implementing an MDS is important and practitioners who may not share this perspective (https://pubmed.ncbi.nlm.nih.gov/19187877/), and wide-spread implementation despite severe accuracy issues of certain elements of the tool without fixing these issues (https://pubmed.ncbi.nlm.nih.gov/27785121/, https://journals.sagepub.com/doi/abs/10.1177/1471301210375337, https://pubmed.ncbi.nlm.nih.gov/15050662/, https://pubmed.ncbi.nlm.nih.gov/16033500/)
--	---

VERSION 1 – AUTHOR RESPONSE

Reviewer-1: Catherine Powell, Institution and Country: University of Bradford, UK		
Questions	Comments	Responses
2	i) BMJ Open requires authors of protocol papers to report dates of the study in the manuscript. There is a registration start date of 09/03/2020, but there is no end date indicated.	Thank you for this helpful comment. This review started from December 2019 and it will end by 31 st December 2020 (Lines 358-359). The date in the protocol indicates when it was registered on Prospero.
	ii) The objective 'To offer transferrable learning and/or utility for an MDS development in countries that do not have an MDS for older people in long term care settings' needs to be clearer in the abstract.	Thank you, this has been rewritten (Lines 35-37).
5	i) Are research ethics (e.g. participant consent, ethics approval) addressed appropriately? From the statement 'The protocol has been submitted to the University of Hertfordshire Ethics Committee.'-does this mean the research is currently under ethical review and not yet approved?	The University of Hertfordshire Ethics Committee has approved this study (ref. HSK/SF/UH/04169). This section has been amended accordingly (Lines 48-49, and 349).
	ii) Stage 3 mentions a series of interviews. The participant consent process needs to be explained.	Thank for this useful observation; this has now been addressed (Lines 282-283).

Reviewer-2: Matthias Hoben, Institution and Country: University of Alberta, Canada		
Questions	Comments	Responses

General observations	1. Clarification and consistency are needed on what the actual focus of the review is. First, there is no clear aim/objectives statement in the abstract.	Thank you for this helpful observation. We have now added aim & objective to the abstract (Lines 35-37).
	2. Clarification about the review aims and objectives If the review wants to address all three, this should be made explicit – in one location of the abstract. If the review wants to address only one or some of them, this should be made clear too – and the other statements should be removed.	We agree the focus of the review requires clarification. This review is interested in what supports the implementation and use of MDS in the care home, its impact on staff working and how it benefits (or not) the care of individual residents. We have rewritten the aims to reflect this focus on the resident level of care and provided examples in the text of possible contexts and outcomes of interest (Lines 35-37; 116-123).
	3. The strengths and limitations box adds a fourth possible review focus: (d) how, why and under what circumstances does the implementation of an MDS lead to benefits (effectiveness, possible side effects and their reasons/mechanisms). Not only is that a focus not yet mentioned, it actually contradicts the statement in the introduction section of the background suggesting that we already know that MDS “help to improve residents’ outcomes and inform care commissioning”.	Thank you, we have rectified the apparent contradiction. The known advantages of standardising assessment and care data for at the population level are clear for policy makers, commissioners, managers and researchers. What is not established is how the uptake and use of MDS improves the resident experience of care, staff working and individual resident outcomes. (Lines 107-111).
	4. Third, additional gaps and possible foci are mentioned in the background (lines 96-100): (e) How does using MDS affect the everyday work and care practices of staff, and (f) how are different priorities of various groups of persons met? Also, (g) what resources are required to sustain use of MDS?	Thank you, please see responses above.
	5. The objective statement and questions at the beginning of the methods section again communicates a slightly different focus. I suggest the authors clearly define what they want to achieve with this review, clearly state it and communicate this consistently throughout the paper (including	We agree. Please see earlier response.

	abstract, strengths/limitations box, main manuscript text).	
Background	i) Clearer definition of 'care homes'	Thank you. This has been addressed in the "background" section (Lines 71-77).
	ii) Lines 83/84: The sentence "Over the next two decades older people likely to need long term care will increase" needs to be fixed. Something is missing (the number of older people?).	Thank you, correction made (Line 89).
	iii) Lines 84/85: I think it should be "is a priority", rather than "are". Determining X is a priority ... and X is "consistent ways to assess and document". Ways is plural but you are describing one priority (determining these ways), rather than multiple priorities.	Thank you, correction made (Lines 90-91).
	iv) Line 87: Either say "the use of MDS has been adopted" (since use is singular) or "MDS have been adopted" (because then MDS is plural). Right now you are mixing plural and singular.	Thank you, correction made throughout the protocol where necessary.
	v) Line 88/89: same issue as in my previous comment: one international consortium (singular) of researchers and practitioners has developed ... or researchers and practitioners (plural) have developed ...	Thank you, corrections have been made as above.
	vi) I think the authors need to discuss in more detail which different versions of an MDS they are interested in. They only mention the interRAI LTCF version. However, the earliest version of an MDS that was implemented system-wide was the RAI-MDS 1.0 (1990 in the US). That version was updated to the version 2.0, which was implemented in the mid 1990s. Then, since 2010 the USA uses the MDS 3.0. In Canada most provinces use the MDS 2.0, some have started to experiment with the interRAI LTCF version. Apart from the RAI family of tools, are the authors interested in other tools?	We use InterRAI as an example of a well-known MDS. We accept that this may have given the wrong impression that the review is not interested in other standardised approaches. This has been reworded with less detail provided about InterRAI but keeping the supporting references for illustrative purposes

	For example PLAISIR – a tool developed and used in the French-speaking part of Canada and used in some regions in Switzerland? Another tool used in Switzerland as an alternative to the RAI-MDS is BESA. There may be more. I think the authors need to be clear on what types of tools they are interested in – and if their interest goes beyond the RAI family of tools, they need to say so and discuss those tools too.	
Methods	a. The authors rightfully state that realist reviews are “theory-driven”. Given that, the discussion on what their exact initial theoretical considerations are is insufficient. Just stating that the realist review will be embedded within the context of a larger systematic review and then list various possible theoretical perspectives that may inform the review (lines 176-184) is not enough. Very likely this very broad and superficial approach of discussing candidate theories may be rooted in the fact that the authors have not clearly and consistently stated the exact aim and objectives of the review.	We have elaborated on how the systematic review searches are the starting point of the review, how the initial programme theory that will guide the scoping and the inclusion of paper
	b. If the objectives and questions stated in the methods section indeed are what will guide the review, this needs to be reflected in the other sections of the protocol – and consistently so. Right now, it appears that the authors’ understanding of the review focus is a bit fussy and varies.	We have addressed this by rewording the aims and objectives
	c. The authors talk about the development of a programme theory – but nowhere do they define exactly what that means. There are excellent definitions of programme theories in the	BMJ Open has published several realist reviews and evaluation protocols that do not include a definition of a programme theory.

	realist methods literature and I highly encourage the authors to clearly define what they mean by 'programme theory', using this literature.	We have included a statement of what characterises programme theory based on Pawson R, Greenhalgh T, Harvey G , et al Realist review—a new method of systematic review designed for complex policy interventions. J Health Serv Res Policy 2005;10(Suppl 1):21–34.doi:10.1258/1355819054308530 We have explained what will inform the initial programme theory at the scoping stage
	d. Included studies: it sounds like the authors will largely focus on studies on care staff experiences with MDS implementation or studies that focus on facility-level implementations and related factors. However, we know that system factors and approaches (legislation, system support, resources, incentives, etc.) are as important for implementation success – especially if a system-wide implementation is the ultimate goal.	The reviewer is correct the realist review is focused on the experience of care home staff and the resident level of care We agree that legislation, incentives and resources directly affect implementation. We do not see this being in contradiction. We reference these wider influences by suggesting that these system influences will affect how staff prioritise resident assessment as part of their care work
	e. The information on how study quality will be determined is insufficient. Just referring to and citing a previous realist review is not enough. Details on how the team will assess whether a paper is 'weak' or 'strong' and whether it is 'relevant' or not are needed.	We have rewritten this section to make it clearer that the criteria for selection are based on whether a paper is likely to contribute to the theory (Lines 231-238).
	f. The entire protocol seems to be driven by the assumption that it is already established that implementing MDS generally has benefits and that it is really just about understanding how to best implement these tools and use their data. However, I think the authors have overlooked important issues in this context:  1) possible negative side effects of implementing and 	That is not the assumption of the review. The review is focused on the consequences of implementing MDS on care home staff and residents. The review will include questions about the cost and consequences for resident care of implementing MDS. For example, impact on staff workload and if some of the claims for accuracy and reliability and how it is used by staff are supported by the evidence.

	using MDS (https://www.ncbi.nlm.nih.gov/pmc/articles/PMC2867498/, https://utpjournals-press.login.ezproxy.library.ualberta.ca/doi/full/10.3138/jcs.50.2.348), 2) differences in perspectives between decision makers/scientists who may think implementing an MDS is important and practitioners who may not share this perspective (https://pubmed.ncbi.nlm.nih.gov/19187877/), and 3) wide-spread implementation despite severe accuracy issues of certain elements of the tool without fixing these issues (https://pubmed.ncbi.nlm.nih.gov/27785121/, https://journals.sagepub.com/doi/abs/10.1177/1471301210375337, https://pubmed.ncbi.nlm.nih.gov/15050662/, https://pubmed.ncbi.nlm.nih.gov/16033500/)	Thank you for the references. These are exactly the literature we are considering.
--	---	---

VERSION 2 – REVIEW

REVIEWER	Matthias Hoben University of Alberta, Canada
REVIEW RETURNED	28-Aug-2020
GENERAL COMMENTS	Thank you the authors for comprehensively addressing all my comments. I recommend publication of the manuscript in its current form and I am excited to see the results of this important realist review.

VERSION 2 – AUTHOR RESPONSE

Queries	Responses
1. Reviewer 1 asked you to clarify the consent process for stage 3. We couldn't locate this information in the revised manuscript. Can you clarify where this is described?	Thank you. Please see Lines 358-370 (Tracked changes version).
2. Study protocol articles do not normally contain conclusions sections. Can you please revise the 'Discussion and Conclusions' heading to 'Discussion'?	Thank you. This has been amended accordingly (Line 404 – Tracked changes version).
3. In principle the methods should be reported in enough detail for others to reproduce your study. Can you work on improving the reporting of your methods? For example, for the literature search what databases will be used? What are the dates of coverage? Can you include a draft of the search strategy as a supplementary file and refer to this in the methods section?	We have included potential search terms and databases used (Lines 191-225 – Tracked Changes version). We have provided a "Supplementary Table S1" outlining potential search terms. We have provided additional text and supporting references to describe the iterative process of a realist review to provide a theory based explanation of how a particular programme works to achieve(or not) its outcomes. Finfgeld-Connett D, Johnson ED. Literature search strategies for conducting knowledge-building and theory-generating qualitative systematic reviews. J Adv Nurs 2013;69(1):194-204. doi: 10.1111/j.1365-2648.2012.06037.x [published Online First: 2012/05/18]